# Skin Phototype and Disease: A Comprehensive Genetic Approach to Pigmentary Traits Pleiotropy Using PRS in the GCAT Cohort

**DOI:** 10.3390/genes14010149

**Published:** 2023-01-05

**Authors:** Xavier Farré, Natalia Blay, Beatriz Cortés, Anna Carreras, Susana Iraola-Guzmán, Rafael de Cid

**Affiliations:** Genomes for Life-GCAT Lab, Germans Trias i Pujol Research Institute (IGTP), 08916 Badalona, Spain

**Keywords:** population genetics, pigmentary traits, phototype, pleiotropy, skin disease, eye disease

## Abstract

Human pigmentation has largely been associated with different disease prevalence among populations, but most of these studies are observational and inconclusive. Known to be genetically determined, pigmentary traits have largely been studied by Genome-Wide Association Study (GWAS), mostly in Caucasian ancestry cohorts from North Europe, identifying robustly, several loci involved in many of the pigmentary traits. Here, we conduct a detailed analysis by GWAS and Polygenic Risk Score (PRS) of 13 pigmentary-related traits in a South European cohort of Caucasian ancestry (n = 20,000). We observed fair phototype strongly associated with non-melanoma skin cancer and other dermatoses and confirmed by PRS-approach the shared genetic basis with skin and eye diseases, such as melanoma (OR = 0.95), non-melanoma skin cancer (OR = 0.93), basal cell carcinoma (OR = 0.97) and darker phototype with vitiligo (OR = 1.02), cataracts (OR = 1.04). Detailed genetic analyses revealed 37 risk loci associated with 10 out of 13 analyzed traits, and 16 genes significantly associated with at least two pigmentary traits. Some of them have been widely reported, such as *MC1R*, *HERC2*, *OCA2*, *TYR*, *TYRP1*, *SLC45A2*, and some novel candidate genes *C1QTNF3*, *LINC02876*, and *C1QTNF3*-*AMACR* have not been reported in the GWAS Catalog, with regulatory potential. These results highlight the importance of the assess phototype as a genetic proxy of skin functionality and disease when evaluating open mixed populations.

## 1. Introduction

Skin, hair, and eye pigmentation are determined by the type and amount of melanin in the basal layer of the epidermis, hair follicles, and the front layers of the iris, respectively [1,2,3,4,5]. In the human skin, the production of melanin takes place through the melanogenesis within the melanocytes [6,7], and is promoted by exposure ultraviolet radiation (UVR), which darkens the skin, the eyes [5] and promotes the development of freckles [8]. Melanin pigmentation protects the skin from the damaging effects of UVR, but exerts a wide variety of functions and harbors different structures and presentations [9], being not only responsible for color, but also found in the light-sensitive tissue of the retina, playing a role in normal vision [10].

Pigmentary traits are genetically determined, heritable [11], and with a polygenic architecture [12]. Constitutive skin color, hair and eye pigmentation, freckles, and the sensibility of the skin (ease suntan/sunburn) influence tanning behaviour in an interplay between genetics and environmental factors [13,14]. In dermatology practice, the structure and functional behavior of the skin is condensed by the phototype, a skin classification not only defined by the skin color, but also including other pigmentary traits (hair and eye color, presence of freckles and sunlight sensitivity).

Several studies have untangled the genetic component of these traits, identifying the main genes associated with several pigmentary traits. *MC1R* was associated with red hair, fair skin, or freckles [15,16]; *OCA2* was previously associated with hair and eye color [17,18]; *SLC24A5*, linked to pigmentation in zebrafish [19], was also confirmed to play an important role in human pigmentation. Specifically, a non-synonymous variant in the third exon of this gene (rs1426654), explains between 25 and 38% of the difference between Europeans and Africans in skin melanin index [19]; IRF4 was associated with skin, hair, and eye pigmentation [20,21] and *TYR* was associated with eye color type [22,23]. However, most of the comprehensive genetics on Caucasian Ancestry for pigmentary traits have been conducted on North European populations [24] or small studies [25,26].

Ancestry background is important because natural selection shapes the overall genetic architecture of skin pigmentation [27]. The latitude hypothesis, explained by favoring vitamin D synthesis in regions far from the equator, is accepted as the main force during ancestral human population spreading [28], but not the only explanation. Recent migrations and cultural factors have brought many people into UVR regimes different from those experienced by their ancestors, and accordingly, exposed them to new disease risks [29]. Interestingly, those genes, while playing an important role in skin pigmentation, are also known to be associated with the prevalence and incidence of skin-colored related diseases, such as melanoma [30,31,32], cutaneous squamous cell carcinoma [33], vitiligo [34], or indirectly, involved in the cancer aggressiveness, where abnormal melanin physiology is involved in the immunomodulation of tumor microenvironment, sustaining melanoma progression and metastasis [35]. Furthermore, modification of the production and packaging of melanin in the skin affects vitamin D levels [36] which are essential for a wide range of physiological processes, including immune function and calcium homeostasis.

In this study, we report a comprehensive analysis of the GCAT cohort, with the identification of phototype associated-diseases in a total of nearly 20,000 subjects and 177 ICD-9 diagnoses, and genome-wide analysis, including structural variants, across 13 pigmentary traits, using polygenic risk scores, in the largest South European population analyzed so far.

## 2. Materials and Methods

Graphical abstract and pipeline including all the material and methods can be found in Appendix A respectively.

### 2.1. Study Population

Participants included in this study belong to the GCAT|Genomes for life cohort, a population-based cohort from Southern Europe (Catalonia, NE Spain) [37]. The GCAT cohort comprises 20,000 volunteers, recruited between 2014 and 2018, from Catalan general population, aged between 40 and 65 years old at the time of recruitment, and 59.16% of women. All the participants completed a detailed, self-reported, baseline questionnaire, including self-reported pigmentary-traits definition and family ethnicity. All participants gave their consent, and all procedures were carried out in accordance with ethical standards. This study was approved by the ethics committee of the Hospital Germans Trias i Pujol (Protocol CEIC PI-13-020).

### 2.2. Phenotype Data Collection

Pigmentary traits were collected for all subjects with a self-report epidemiological questionnaire, during the recruitment (including 19,205 valid responses). These data included single pigmentation-related phenotypes, comprising: three binary phenotypes for hair color, including red hair (red hair as cases, and blond, brown and black hair as controls), blond hair (blond hair as cases, and brown and black hair as controls), and brown hair (brown hair as cases and black hair as controls); eye color (i.e., light blue or green, blue or green, hazel, brown and dark brown); constitutive skin color (i.e., black, dark, medium, white and very white measured in the upper-inner arm); freckles (i.e., no, occasional, few, some and abundant); ease sunburn (i.e., never, rarely, sometimes, moderately and easily); ease suntan (i.e., never, slightly, moderate and strongly); as well as sun habits, such as the use of sunscreen in the exterior (i.e., never or rarely, sometimes, almost all the time, always or not in the exterior during sunny moments); the use of sunscreen when suntan (i.e., never or rarely, sometimes, almost all the time, always or not in the exterior during sunny moments) and weekly hours spent outside (Appendix A). For the phototype study, we included the Fitzpatrick scale, which is a phototype variable calculated from the aggregation of self-reported variables based on the widely used Fitzpatrick skin phototype classification [38], including information about skin color, hair color, eye color, freckles, ethnicity, sunburn, and suntan. This information was obtained from a questionnaire where a score was calculated (Appendix A) and transformed into the six phototype categories (from I to VI) corresponding to Fitzpatrick skin phototype classification (see description in Appendix A). A second variable, phototype score, based on the raw score from the questionnaire was also included in the study for analysis purposes. This score ranged from 0 to 86 in our study, where the higher scores represented the darker skins with higher sunlight tolerance. A detailed description of all included variables can be found in Appendix A.

### 2.3. Genetic Variants

Genetic analysis was conducted in a subsample of the GCAT cohort (GCATcore) that is fully genotyped by SNP-array and imputation. This subsample included 4988 unrelated participants with Iberian ethnicity, determined by self-described ethnicity, and supervised ancestry inference by Principal Components Analysis (PCA), as previously described by Galván Femenía et al. [39]. Genotyping was completed using the Infinium Expanded Multi-Ethnic Genotyping Array (MEGAEX) (ILLUMINA, San Diego, CA, USA). Then, we imputed both SNVs and SVs, with IMPUTE2, and using the GCAT|Panel as reference [40], an ancestry-geographically matched panel generated by whole-genome sequencing. The final core set included a total of 10,216,971 variants with MAF ≥ 0.01 and INFO score ≥ 0.7 (21,620 SVs, and 10,195,351 SNVs and small indels). All data are available at EGA (Study ID EGAS00001003018).

### 2.4. Electronic Health Records Disease Prevalence and Phototype

Medical diagnoses of the whole cohort were obtained from Electronic Health Records (EHR) of the Public Healthcare System of Catalonia. All diagnostic information (2010-current) is linked to each participant, within the framework of the PADRIS Program as provided by the Agency of Health Quality and Assessment of Catalonia (AQuAS) [41]. We selected all ICD-9 minimum basic data set (MBDS) diagnoses from the period 2012 to 2017, grouped them into a three digits’ level, and selected the ones with a prevalence of at least 1%. A total of 177 diagnoses were finally included in the analysis. Then, a regression analysis was conducted for each disease with the Fitzpatrick scale (phototypes I-VI), correcting by age and gender. A Bonferroni correction by the number of ICD-9 chapters (n = 15) was applied, accounting for redundancy. Results are reported as odds ratios (OR) with 95% confidence intervals (CI) and *p*-value (*p*).

### 2.5. Genome-Wide Association Analysis of Phototype

Independent genome-wide association analysis (GWAS) for the 13 pigmentary-related traits (as three binary and ten continuous variables). Red hair was analyzed separately from blond and brown color due to its unique genetic architecture. Final analysis included a total of 10,195,351 SNVs and small indels, and 21,620 SVs with a Minor Allele Frequency (MAF) ≥ 0.01 and INFO score ≥ 0.7. Linear and logistic regression models were used for continuous and binary traits respectively using PLINK (version 2.0) [42], assuming an additive genetic model and including age and sex as covariates, as well as the first 10 PCs to control the potential population stratification. Standard genome-wide threshold (*p* < 5 × 10^−8^) was kept for significance. SNPs were functionally annotated and mapped to the closest gene with SnpEff [40]. Manhattan and quantile-quantile (QQ) plots were generated to explore the results, as well as a 3D Manhattan representation that provides a visual representation of the common GWAS hits between the different traits. Regional Manhattan plots for the associated SVs were represented, including information from the genes in that region obtained from GENCODE (v40) [43]. Additionally, the association results were analyzed using MAGMA [44], a gene-based approach that aggregates multiple genetic markers into genes to determine their joint effect.

Then, for each GWAS, independent genomic loci were identified as described in Watanabe et al. [45], using GCAT|panel [40] as the reference panel to compute the linkage disequilibrium (LD) measure. First, we defined independent significant SNPs (PLINK software, p1 = 5 × 10^−8^, linkage disequilibrium threshold r2 = 0.6, and physical distance threshold for clumping 1000 kb), and lead SNPs (PLINK software, p1 = 5 × 10^−8^, r2 = 0.1, and distance 1000 kb). Then, LD blocks that were closer than 250 kb were merged into genomic trait-associated loci.

To further elucidate the biological mechanisms underlying pigmentary traits, we performed a functional enrichment analysis using WebGestalt [46], which tests for an overrepresentation of the genes resulting from the gene-based analysis performed with MAGMA, in specific GO terms, pathways, and disease-associated gene sets.

### 2.6. Trait Heritability and Pairwise Genetic Correlations

We estimated the SNP heritability of each individual trait (h2), and computed pairwise genetic correlations, using LDSC (v1.0.1) [47] with the precalculated LD scores for 1000 Genomes phase 3 for European population. For binary traits, SNP heritability and genetic correlations were computed at the liability scale, assuming that the GCAT sample prevalence is equal to the Spanish population prevalence.

### 2.7. EHR Phenotype-Wide Association

A Phenotype Wide association study (PheWAS) was conducted with former identified lead SNPs using the publicly available data PheWeb [48] that includes GWAS results for 1419 EHR-derived PheWAS codes in approximately 400,000 White British participants of the UK Biobank [49]. Effect sizes directions from the PheWAS analysis were referenced to the same effect allele as the one in our GWAS. For strand ambiguous SNPs, allele frequencies were checked to reference the data to the same effect allele, excluding those SNPs in which the allele frequency ranged from 0.4 to 0.6.

### 2.8. Polygenic Risk Scores Association

Polygenic risk scores (PRS) were derived for selected relevant skin and eye diseases, based on disease-phototype association, GWAS results, and available genetic data from public databases. Here we present data for three skin cancer phenotypes (including melanoma, basal cell carcinoma, and non-melanoma skin cancer), two eye diseases (glaucoma and cataracts), two immune-related dermatoses (psoriasis and vitiligo) and metabolic diseases (essential hypertension and overweight, obesity and other hyperalimentation). We used weights for relevant diseases from the PGS Catalog [50] giving priority to those polygenic scores in which the source of variant associations (GWAS) was from European ancestry (Appendix A). We harmonized the data by removing strand ambiguous SNPs and discordant SNP alleles. Then, polygenic risk scores for each individual were computed using PLINK v.1.9 [42]. Associations between pigmentary traits and polygenic risk scores were assessed using linear or logistic regression, accounting for the age and sex of the participants, as well as the first 10 PCs to control for population stratification. Results are reported as odds ratios (OR) with 95% confidence intervals (CI), and *p*-value (*p*).

## 3. Results

### 3.1. Systematic Search on Clinical Data (EHR) Identifies Diseases Associated with Phototype in the GCAT Cohort

Among the selected 177 EHR ICD-9 diagnoses, seven diseases were significantly associated with a fair Fitzpatrick scale phototype after Bonferroni correction, and eight were nominally associated (Appendix A). Among the significantly associated diagnoses, *Diseases of the skin, and subcutaneous tissue* and *Neoplasms* were the two most represented ICD-9 chapters (71%). The most significantly associated diseases were: *Other and unspecified malignant neoplasm of skin* (ICD-9 code 173, OR = 0.60, CI = 0.5–0.71, *p* = 1.97 × 10^−9^), and *Other dermatoses* (ICD-9 code 702, OR = 0.79, CI = 0.7–0.88, *p* = 1.67 × 10^−5^). Other prevalent diseases included: *Benign neoplasm of skin* (ICD-9 code 216), *Contact dermatitis and other eczema* (ICD-9 code 692), and *Erythematous conditions* (ICD-9 code 695). Moreover, we found two common conditions associated with fair phototype: *Overweight, obesity and other hyperalimentation* (ICD-9 code 278) and *Essential hypertension* (ICD-9 code 401).

### 3.2. GWAS Analysis Identifies Thousand Variants Associated with 10 Pigmentary Traits in the GCAT Cohort

We carried out a GWAS across > 10M variants with a MAF ≥ 0.01 and INFO score ≥ 0.7 for each individual pigmentary trait, including SNVs and SVs (Appendix A). Single-variant analysis identified 3562 associations at 1515 SNPs in 10 pigmentary traits (red hair, blond hair, brown hair, eye color, freckles, ease sunburn, ease suntan, skin color phototype score and Fitzpatrick scale), the other three (sunscreen in exterior, sunscreen when suntan and weekly hours spent in the exterior) did not show any significant association. The 1515 associated SNPs were grouped into 257 independent SNPs and 99 unique lead SNPs grouped into 37 genomic risk loci, mapping to 48 genes (Appendix A). A summary of genome-wide significant variants and their associated loci is shown in Table 1. Detailed results regarding SVs are described later in Section 3.5. Twelve out of the 48 genes corresponding to six loci, were pleiotropic, associated with more than one pigmentary trait (*SLC45A2*, *HERC2*, *OCA2*, *IRF4*, *MC1R*, *ADAMTS12*, *RXFP3*, *TYR*, *TYRP1*, *NOX4*, *AMACR*, *EXOC2*). Noteworthy, *SLC45A2*, a critical determinant of skin and eye pigmentation, was the most pleiotropic gene, associated with nine pigmentary traits.

GWAS results were represented in a 3D Manhattan plot to highlight the pleiotropic nature of associated regions (Figure 1). *SLC45A2* (lead rs16891982) was associated with nine traits (blond hair, brown hair, eye color, ease sunburn, ease suntan, Fitzpatrick scale, phototype score, skin color and freckles); *IRF4* (lead rs12203592) show association with six traits (brown hair, eye color, ease sunburn, ease suntan, skin color and freckles); *HERC2* (lead rs12913832) was associated with seven traits (blond hair, brown hair, ease sunburn, ease suntan, Fitzpatrick scale, phototype score and skin color) and *MC1R* (lead rs1805007) presented association with four traits (red hair, ease suntan, Fitzpatrick scale and phototype score). Pleiotropy, as expected, was evident for the aggregated variables of phototype, which include dependent variables, but it was also higher for skin color, suntan, sunburn, and brown hair; and lower in red and blond hair color and eye color (Figure 2). Since there was a gene overlapping in the 37 genomic risk loci, we refined these results using a gene-based analysis for further steps (Section 3.4).

### 3.3. Pigmentary Traits Correlation Are Partly Explained by Common Genetics

SNP heritability of the pigmentary traits, as well as genetic correlations between them, were measured using Linkage Disequilibrium Score Regression (LDSC) technique. This allowed us to have an overview of the phenotypic variance explained by the SNPs, as well as to obtain a measure of the shared genetic basis between these traits. The higher heritability was observed for hair color (e.g., brown hair: h2 = 0.42, se: 0.189; blond hair: h2 = 0.67, se: 0.381), phototype score (h2 = 0.28, ee: 0.112), and Fitzpatrick scale (h2 = 0.20, se: 0.197). Heritability estimates for red hair (h2 = 1.916, se: 2.001) were unreliable due to the low number of cases.

All the traits showed a high genetic correlation (Appendix A). As expected, the analysis showed a high correlation between the two phototype measures; Fitzpatrick scale and phototype score (rg: 0.96, se: 0.08, *p*: 1.1 × 10^−^^32^). Furthermore, phototype measures correlated with eye and hair measures and ease sunburn were highly correlated with eye color and phototype measures. Regarding habits, sunscreen use in exterior was correlated with sunscreen when suntan and inversely sunscreen when suntan and weekly hours in the exterior. Red hair was highly correlated with sunscreen in, sunscreen when suntan and ease sunburn, However, the low number of red hair cases in the cohort made an accurate estimation difficult (Appendix A). Among all, four traits remained significant; a negative correlation among ease of sunburn and phototype score (rg: −1.15, se: 0.25, *p*: 2.88 × 10^−^^6^), ease of sunburn and Fitzpatrick scale (rg: −1.43, se: 0.52, *p*: 6.08 × 10^−^^3^); and among brown hair and light hair (meaning brown versus black trait) with phototype score (rg: −0.85, se: 0.29, *p*: 3.89 × 10^−^^3^).

### 3.4. Genes Containing SNPs Associated with Pigmentary Traits, Show an Overrepresentation of Pigmentary Metabolic Processes, Skin, and Eye-Related Diseases

To further characterize the functional impact of the SNPs identified in the GWAS analysis, we performed a gene-based analysis, with aggregated SNPs into single genes. Then, we tested the association between genes and pigmentary traits using MAGMA software [42] (Figure 3A, Appendix A). After Bonferroni correction, 22 genes had a significant association with any of the pigmentary traits. Two genes, *HERC2* and *SLC45A2*, out of the 22 showed associations with skin, hair, eye, and phototype. *DPEP1* and *SPATA33* were associated with skin, red hair, and phototype. *MC1R*, *TYR*, *ADAMTS12*, *AMACR*, *C1QTNF3*, *CHMP1A*, *GAS8*, *C1QTNF3-AMACR*, *RXFP3*, and *SPIRE2* were associated with skin and phototype, and *OCA2* and *TYRP1* were associated with eye and phototype traits. Six identified genes did not show an association with phototype, being exclusively associated with one trait: LINC02876, *DEF8*, *VPS9D1*, and *ZNF276* with red hair, *SLC25A37* associated with sunscreen, and TRPS1 with skin color. *C1QTNF3-AMACR*, *C1QTNF3* and LINC02876 were not previously reported as pigmentary traits in GWAS Catalog.

Functional enrichment analysis using WebGestalt [46] revealed an overrepresentation of genes involved in the melanogenesis pathway, tyrosine metabolism, monooxygenase, and glycolipid binding functions. Moreover, this analysis revealed an enrichment of the genes associated with diseases such as albinism, melanoma, basal cell carcinoma, vitiligo, or visual impairment, and with Hermansky–Pudlak Syndrome (Figure 3B, Appendix A).

### 3.5. Analysis of Structural Variants Identified Small Intronic Deletions as the Most Common Structural Variation Associated with Eight Pigmentary Traits

The use of the GCAT|Panel as the reference panel for imputation allowed us to impute not only SNV information, but also SVs to investigate functional consequences. A total of 15 SVs showed a suggestive significant association with at least a pigmentary trait (Appendix A), and eight of them surpassed the genome-wide significance threshold. All identified SVs were located in already identified risk loci and did not lead to an association. For the impact interpretation of SVs, mapping genes were considered based on closest gene to SVs, and this could have generated some discrepancy in the cited genes. Red hair and phototype (both phototype score and Fitzpatrick scale) harbors the higher number of significant loci; being the more significant, an intronic deletion in *SPIRE2* (16:89896057:G:<DEL>) associatedwith red hair (OR = 25.82, CI = 13.79–48.35, *p* = 3 × 10^−24^), phototype score (OR = 1.72 × 10^−3^, CI = 3.18 × 10^−4^–9.39 × 10^−3^, *p* = 2.04 × 10^−13^), Fitzpatrick scale (OR = 0.73, CI = 0.67–0.8, *p* = 2.58 × 10^−12^) and also present in ease suntan (OR = 0.62, CI = 0.55–0.69, *p* = 3.84 × 10^−18^). Red hair was also associated with three other intronic deletions in *GAS8*, *CHMP1A*, and *CPNE7*. Other significant associated loci were two small intronic deletions associated with eye color in *HERC2* (15:28473597:A:<DEL>) (OR = 1.51, CI = 1.39– 1.65, *p* = 2 × 10^−21^) and *LURAP1L-AS1* (9:12756780:C:<DEL>) OR = 1.14, CI = 1.09–1.19, *p* = 2.56 × 10^−9^), both also associated with phototype score. Moreover, two intronic deletions associated with skin color; a small deletion in *BNC2* (9:16876769:G:<DEL>) (OR = 0.91, CI = 0.88–0.94, *p* = 1.05 × 10^−8^), and a middle-deletion in *SLC45A2* (5:33973937:G:<midDEL>) (OR = 0.89, CI = 0.86–0.92, *p* = 1.6 × 10^−12^). In summary, most of the identified SV were deletions (13) and, only suggestively, one insertion (Alu) in chromosome 6 (*DST*) and one translocation in chromosome 7 (*ZNF92*) were observed (Figure 4). The majority of the SVs mapped onto intronic or intergenic regions (11 out of 15 SVs) (Appendix A).

### 3.6. PheWAS Analysis Confirmed the Pleiotropy among Pigmentary Traits Genes and Related Diseases

To assess whether the lead SNPs associated with pigmentary traits identified in our analysis were previously associated with diseases, we performed a Phenotype Wide Association Study (PheWAS) analysis with these SNPs and using a PheWeb dataset that contains the results of associations carried out with the UK Biobank data, as described in the Material and Methods section. This analysis revealed that 17 out of the 99 SNPs had at least an association surpassing the suggestive significance threshold in the UKbiobank (*p* = 1 × 10^−5^). Amongst the diseases, those belonging to the neoplasm and dermatologic categories were the most frequent ones (Appendix A). Highly pleiotropic variants mapped onto three fundamental pigmentary genes *IRF4*, *HERC2*, *TYR*; rs12203592, which is an intronic variant in *IRF4* gene, was significantly associated with six of the pigmentary traits in our analysis (brown hair, ease sunburn, ease suntan, eye color, freckles, and skin color), and with skin cancer and actinic keratosis. The allele T is associated with fair skin pigmentation, and increasing ease of sunburn, but also increased the risk of melanoma or actinic keratosis (Figure 5A); rs12913832, an intronic variant in *HERC2* gene, significantly associated with eight of the pigmentary traits (blond hair, brown hair, ease sunburn, ease suntan, Fitzpatrick scale, phototype score and skin color), and diseases affecting the eyes, such as cataract or glaucoma. The allele G, associated with fair skin or eye color, decreased the risk of glaucoma and cataract (Figure 5B); and rs1126809, a missense variant in *TYR* gene significantly associated with three pigmentary traits (ease sunburn, Fitzpatrick scale and phototype score), and with neoplasms and bipolar disorder. Here, the allele A is associated with fair skin color, increased risk of ease of sunburn, and risk of skin cancer and bipolar disorder (Figure 5C).

### 3.7. Polygenic Risk Scores Confirmed the Association of Pigmentary Traits and Eye Diseases

To further confirm the shared genetics between pigmentary traits and diseases, we assessed the association between PRSs for these diseases with pigmentary traits. We observed significant associations between pigmentary traits and melanoma, non-melanoma skin cancer, basal cell carcinoma, vitiligo, cataracts, and overweight, obesity and hyperalimentation. Fitzpatrick scale, ease sunburn, ease suntan, sky color and freckles, showed the same association pattern for melanoma, non-melanoma skin cancer, vand basal cell carcinoma, and the opposite for vitiligo and cataracts. No clear pattern was observed for overweight. Individuals with a fair phototype present a higher PRS for melanoma, non-melanoma skin cancer and basal cell carcinoma (Figure 6A–C), and a trend for increased risk overweight, obesity and hyperalimentation (Figure 6D). Individuals with dark phototype present a higher PRS for vitiligo and cataracts (Figure 6E,F). No significant associations were observed for glaucoma, psoriasis, and essential hypertension. Individuals that had more skin sensitivity (ease sunburn) had an increased risk for melanoma, non-melanoma skin cancer, basal cell carcinoma, and overweight. An inverse association with ease suntan was also observed for all of them.

## 4. Discussion

The fact that differences among individuals and populations are mainly due to the genetic variation component of the human genome is also extensible to pigmentary traits [51], making it plausible that certain pathologies, previously associated to different ancestries, found their cause in the onset of a single mutation, a deletion, or an inversion that is fixed over the time together with pigmentary traits. Most GWAS of pigmentary traits, have been based on North European cohorts [52,53,54], suggesting the need for more studies in a geographically distinct population. In this context, we presented a GWAS study of 13 pigmentary traits (constitutive/facultative pigmentation, and behavioral related traits) in a South European cohort from Catalonia (NE of Spain).

Our study confirmed previous findings in North European populations. Fair phototypes are significantly at more risk of developing skin-related diseases, such as neoplasm and dermatosis, and to a lesser extent, benign neoplasms, dermatitis, eczema, and erythema. This predisposition has largely been related to the pheomelanin present in keratinocytes of fair phototypes, with a truthful mechanistic explanation by (i) an overall increase in DNA damage that overloads the DNA repair machinery [55], (ii) release of reactive oxygen species (ROS) that induce more DNA damage [56] and (iii) immunosuppressive effect of UVR [57].

In addition, our study highlighted that the fair skin phototype was also associated vitiligo, cataracts, glaucoma, and psoriasis, and with two common conditions non primarily related with skin color: obesity and essential hypertension. Obesity represents a significant health problem and together with hypertension significantly increases the risks of suffering from other major diseases affecting vital organs as heart, brain, kidney, and other diseases. Several authors suggested that deficiency of vitamin D, may be linked to obesity, hypertension, hyperlipidemia, and diabetes, increasing metabolic and cardiovascular risk [58,59], thus relating these conditions to functional ability to synthetize vitamin D. This hypothesis reinforces the role of geographical location and lifestyle factors (diet, physical activity, clothes, etc.) related with the association observed, but also be conditioned by the skin phototype, and the close relation with vitamin D metabolism. Preliminary genetic analysis of vitamin D shown a genetically determined basis in the PheWas analysis, but no able to show its association with phototype in our cohort From our data, we cannot extrapolate such a conclusion, and more refined attention is needed to identify the exact mechanisms mediating this association.

When looking at genes, we observed 16 genes associated with one or more pigmentary traits and diseases. Six of them were largely associated with melanogenesis and pigmentary traits, in different ethnic groups (*MC1R*, *HERC2*, *OCA2*, *TYR*, *TYRP1*, *SLC45A2*) [39,60,61]; other have been previously reported to be related with: hair color, hair measurement, suntan, sunburn, but also with non-melanoma skin carcinoma, cutaneous melanoma, pigmentary retinopathy and oculocutaneous albinism, (*ADAMTS12*, *AMACR*, *CHMP1A*, *DPEP1*, *GAS8*, *RXFP3*, *SPATA33* and *SPIRE2)* [24]. Our analysis identified for the first time, to our knowledge, the association of novel candidate genes *C1QTNF3, C1QTNF3*-*AMACR* and *LINC02876* to the studied phenotype. *C1QTNF3*-*AMACR* is a conjoined gene [62] located in a highly variable genomic region, mapping on the reverse strand of chromosome 5, which generates a 9-exon alternative readthrough transcript, classified as a non-sense mediated decay mRNA (NMD). It is originated from parent genes *C1QTNF3* (Complement C1q Tumor Necrosis Factor-Related Protein 3) and *AMACR* (α-methylacyl-CoA racemase, previously associated with hair color) the three of them being associated with skin color and phototype, in the present study. NMDs functions not only as a quality control mechanism targeting aberrant mRNAs containing a premature termination codon but also as a posttranscriptional gene regulation mechanism targeting numerous physiological mRNAs [63], revealing a putative complex hotspot locus for fine regulation of phototype and pigmentary traits of especial interest for further studies. *LINC02876* located in chromosome 17, also known as *LINC02876*, is a long non-coding RNA, associated with red hair in our study, a trait with a low prevalence in our cohort, making necessary further validation. Some of the identified lead SNPs, mapping *IRF4*, *HERC2*, and *TYR*, were also found to be associated with melanoma or other skin diseases in the UKBiobank. Mutations in the *MC1R* gene lead to fair skin and red hair in humans, which is also seen with inactivating human *POMC* gene mutations. The powerful effects of proopiomelanocortine (POMC) and probably Corticotropin-releasing hormone (CRH) on the skin pigmentary, immune, and adnexal systems are consistent with stress-neutralizing activity addressed at maintaining skin integrity to restrict disruptions of internal homeostasis. This CRH/POMC skin system appears to generate a function analogous to the HPA axis that in the skin is expressed as a highly localized response which neutralizes noxious stimuli and attendant immune reactions [64,65]. Alterations of this system linked to phototype could lead to a different immune status related to some observational associations with diseases of immune component as psoriasis.

While most of the GWAS studies have been restricted to SNVs and small indels, our study incorporated a large number of structural variants in the analysis [49]. GWAS analysis identified 15 SVs, mostly small deletions, that were at least at the suggestive significance level for any of the pigmentary traits, mostly located in intronic or intergenic regions of genes known to influence pigmentary traits (*SLC45A2*, *TYR*, *OCA2* and *HERC2*), *CHMP1A* and *GAS8*, both close to *MC1R*, associated with red hair [17] and hair color [66], respectively, or potentially involved with new candidate genes, as *BNC2*, a transmission factor of keratinocytes, and *SPIRE2* involved in the transport of intracellular vesicles [67,68], *SEMA6D*, a member of the semaphorin family showing a suggestive association with hair greying [69] and *CPNE7* associated with skin tanning [54]. Unfortunately, most of the observed SV did not map onto any known functional regulatory element to attribute a functional impact. However, some identified regions, as the observed SV mapping at *LURAP1L-AS1*, were interestingly linked to melanin genes. *LURAP1L-AS1* is an antisense RNAs, a unique DNA transcript, small, noncoding, and diffusible molecule that complements mRNA, and are recognized as intracellular gene regulators. *LURAP1L-AS1* gene overlaps *TYRP1*, involved in eumelanin synthesis [70,71], and suggest a role in *TYRP1* transcript regulation and putative melanin dysregulation.

Complementary to single SNP analysis, gene-based analyses provided additional power due to the aggregate effect of multiple SNPs being larger than that of individual SNPs. The functional enrichment analysis revealed an overrepresentation of defective pigmentation diseases involving altered pigment metabolic processes (albinism, lentigines), involved in melanogenesis, or tyrosine metabolism, involving monooygenase activity or the glycolipid binding functions, probably related to melanin, and bile acids biosynthesis. Glycosphingolipid metabolism is enriched in the gene dataset, and glycosphingolipids have been reported essential for a protein-sorting step in the Golgi complex. It has also been suggested that the Hermansky–Pudlak syndrome, a rare disease with associated albinism, has its origin in the altered lipid metabolism [72]. Indeed, a strong reduction in melanin production and reduced tyrosinase activity in melanocytes of both light and dark skin type origin have been reported in cell cultured human melanocytes with lipid metabolism inhibitors. Other enriched pathways include well-reported antioxidant molecules, glutathione (GSH). Cellular antioxidant systems, such as the thioredoxin (Trx) and glutathione (GSH) systems, function to reduce oxidative stress. Therefore, the antioxidant defense systems play a vital role in maintaining an optimal redox balance in melanocytes by quenching ROS and protecting against oxidative stress, excessive melanogenesis and photo-damaged skin. Furthermore, our results show an overrepresentation of genes linked to key cellular components as pigment granules, or ESCRT complex, that are required for Tyrp1 transport from early endosomes to the melanosome limiting membrane [73]. In addition, it is worth noting that out identified genes are enriched in genes involved in several eye diseases, such as photofobia, myopia, hyperopia, retinal degeneration, or fovea hypoplasia. Foveal hypoplasia is an ocular abnormality that is typical associated with other ocular findings as nystagmus, aniridia, cataract, and skin hypopigmentation.

Pleiotropy analysis confirmed shared genetics for pigmentary traits, melanoma, and non-melanoma skin cancer. The results suggest a higher risk of skin cancer in individuals with a fair phototype, consequent with a protective effect from UVR damage to the skin and the ability of melanin to absorb from 50% to 75% of UVR, but also could be related to the described pathological role of melanin, linking a defective secretion mechanism to an increase in tumor severity [35]. In addition, vitiligo, cataracts, and overweight shared genetics with pigmentary traits, being darker phototypes at higher risk of vitiligo, cataracts and overweight. In the case of vitiligo, this result supports the observed different prevalence across latitudes, with a higher prevalence in populations with darker skin color [74]. The mechanism of vitiligo is still unclear but an indirect effect of the deficient levels of vitamin-D production could alter immune status and thus it could be involved in autoimmune dysregulation. Another example of immune disease for which a deficit of vitamin-D has been largely reported is psoriasis [75]; however, we did not observe any phototype association in our dataset when analyzed. Regarding increased risk of cataracts in individuals having darker phototypes, the results confirmed the previous epidemiological observational that showed an association between dark eyes and cataracts [76]. Plausible mechanisms could be related to the increase in the temperature of the lens when dispersing the UVR [77]. However, reversal association could not be discarded, since cataracts are caused by the degradation of proteins and fibers of the lens, it may be linked to behavioral habits, as spending too much time in the sun without sunglasses, which is more common in individuals with a darkness phototype based on a self-perception of security. In the case of overweight, the association between darker color, ease sunburn, and weekly hours spent in the exterior was associated with no apparent mechanistic explanation but it was linked to a more complex behavior. Indeed, from the 13 included pigmentary traits, three phenotypes (“sunscreen in the exterior”, “sunscreen when suntanning”, “weekly hours expend in the exterior”) did not show any gene association. These variables are defined as behavioral habits, more related to social and self-risk-perception than to a direct biological on skin function, and probably are prone to reversal causation in the association analysis. Obesity is responsible for changes in skin barrier function, wound healing, microcirculation, and subcutaneous fat. Moreover, obesity is implicated in a large number of dermatological conditions (see review in Yosipotvitch et al. [78]). Interestingly, Sultan et al. reported that after UV injury, the induction of the melanogenic phase activates fatty acid biosynthesis results in the formation of triacylglicerids, suggesting that an alteration of the fatty acid metabolism being involved in a variety of cutaneous diseases manifesting hyper pigmentary phenotype [79].

Our results provide a comprehensive analysis of the genetic basis of pigmentary traits in a large South European population and offer a landscape of their genetic basis for a better understanding of human pigmentation pleiotropy based on the common genetic background to some previously observed associations. Furthermore, our findings support the importance of incorporating population information to adapt personalized care approaches to consider differential risk in general population, being of great importance in modern mixed populations. It is well accepted that natural selection shaped the skin pigmentation phenotype, resulting, in a gradient of variation across populations, directly correlated with absolute latitude [13,14,80]. Understanding the genetic basis of adaptation of skin color in different populations have many implications for human evolution and medicine [13]. However, the healthy bias of the GCAT cohort is a limitation that could preclude the extrapolation to the general population Among other identified limitations of the study, we need to consider that the cohort, even if there is a wide distribution of phototypes, there may be some underrepresented ancestries. Moreover, self-reported questionnaires could introduce some bias, due to self-perception of ancestry or social pressure. The reduced number of individuals with less represented phenotypes in the population, such as red hair color, may bias some results and should be considered with caution. Lastly, the partial genetic analysis conducted, where a complete analysis including all the cohort and more detailed genetic data, such as rare variants coming from WGS or WES, would be needed to identify more specific private population effects. Phototype could be a tool to define preventing health strategies in general population, however since reversal association is possible, further detailed studies are needed to understand the role of pigmentation traits in health and disease.

## Figures and Tables

**Figure 1 genes-14-00149-f001:**
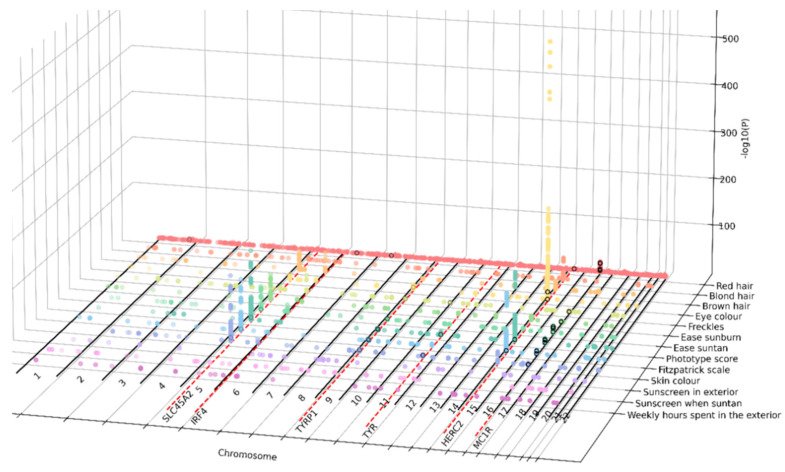
Combined GWAS results for the 13 pigmentary traits highlight their pleiotropic nature. Three-dimensional Manhattan plot depicting in the X axis the chromosomal positions of the genome-wide significant associations; in the Y axis, pigmentary traits represented by colors; and in the Z axis, the level of significance of the association, expressed as −log10 (*p*-value). Colored dots represent the lead SNPs associated to one or more pigmentary traits. Black circles represent SVs, overlapping with significant GWAS hits. Red-dotted lines represent the mapping of genome-wide significant associated SNPs with several pigmentation traits mapping to *SLC45A2*, *IRF4*, *TYR*, *HERC2*, and *MC1R* genes.

**Figure 2 genes-14-00149-f002:**
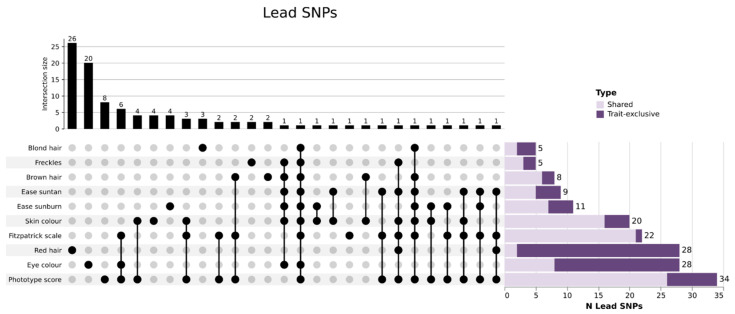
Combination analysis of leading SNPs per pigmentary trait. On top, bar-plot showing the total count of shared or trait-exclusive leading SNPs for each trait or combination of traits. Bottom-left, diagram representing the distribution of shared/trait-exclusive leading SNPs (black dots). Bottom-right, bar-plot representing the total count of leading SNPs both shared (light purple) and trait-exclusive (dark purple) per trait. The combination analysis indicates that red hair and eye color have the higher number of trait-exclusive leading SNPs, highlighting the unique genetic architecture of these two pigmentary traits compared with the other traits analyzed.

**Figure 3 genes-14-00149-f003:**
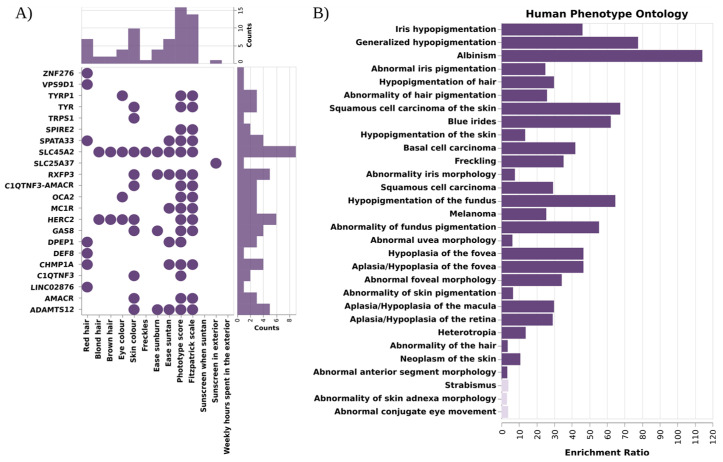
Results of the gene-based analysis and functional enrichment. (**A**) Scheme depicting the total count of associations (dots) per gene (bottom bar plot) and trait (right-hand bar plot) identified by gene-based analysis with MAGMA software. *SLC45A2* was the most pleiotropic gene, being associated with 9 pigmentary traits; on the other hand, phototype traits were the ones with more associated genes (**B**) Functional enrichment analysis of the 22 aggregated genes with WebGestalt. Bar plot depicting enrichment ratio in the X axis, and the different Human Phenotype Ontologies in the Y axis. Dark color indicates that pass the FDR threshold. Pigmentary traits and diseases are significantly enriched in these genes.

**Figure 4 genes-14-00149-f004:**
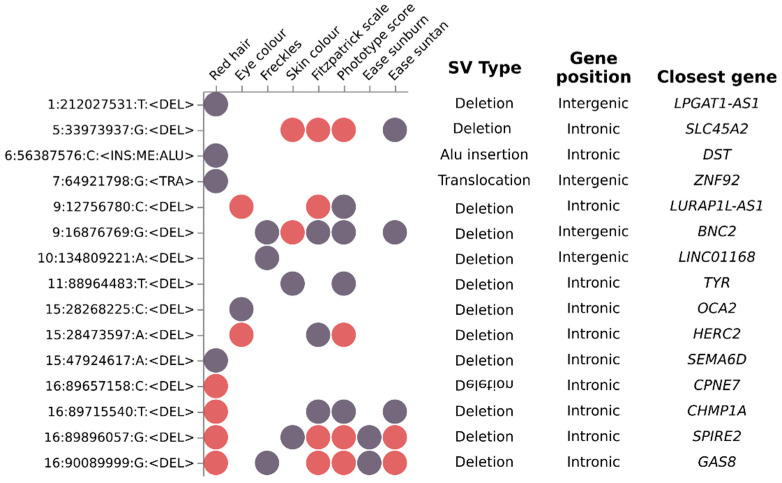
Genome-wide analysis identified structural variants significantly associated with a subset of pigmentary traits. Dot-plot depicting the suggestive (purple dots) and significant (red dots) association of fifteen SVs (Y axis) with a subset of eight pigmentary traits (X axis). Additional SV information are shown in SV type (deletion, Alu insertion and translocation), Gene position and Closest gene columns. Most Identified SV were deletions located in intronic regions.

**Figure 5 genes-14-00149-f005:**
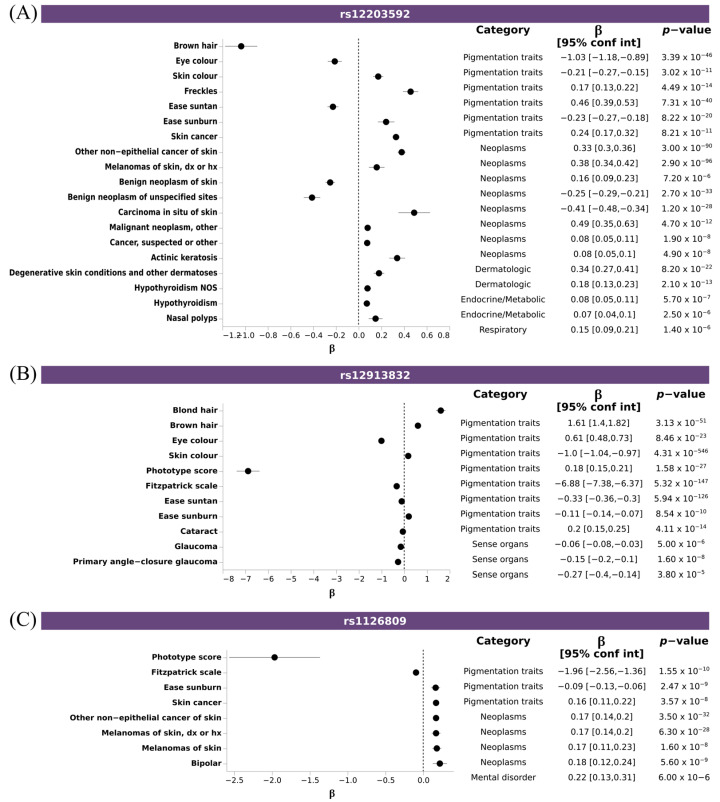
Pleiotropic SNPs associated with pigmentary traits and diseases. Forest plot depicting the effect size of disease-trait association of selected SNPS: (**A**) rs12203592 (*IRF4*), (**B**) rs12913832 (*HERC2*), and (**C**) rs1126809 (*TYR*). X axis indicated the effect size (β) for any trait reaching suggestive threshold (Y axis). Additional information about trait category, β, CI and *p*-value is shown.

**Figure 6 genes-14-00149-f006:**
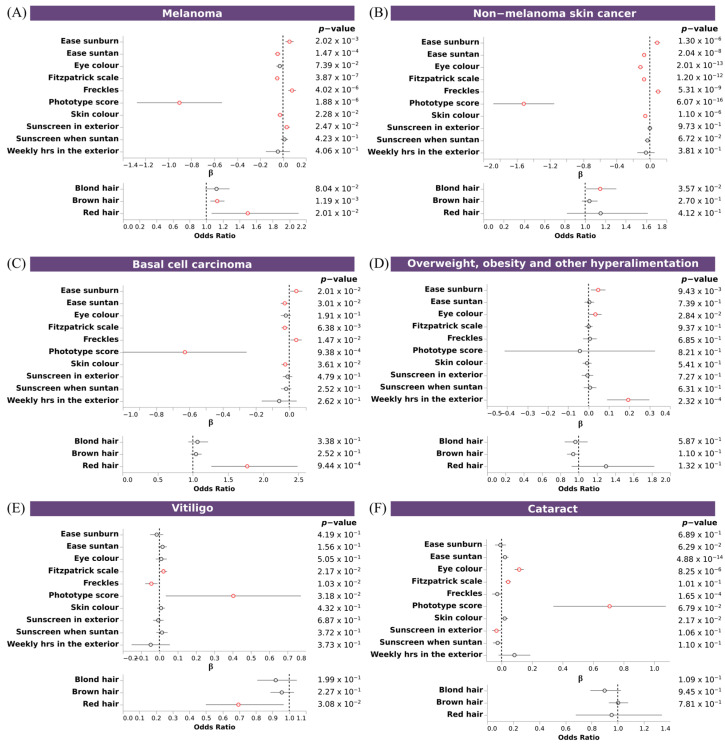
Association of PRS of skin and sense organs diseases with pigmentary traits. Forest plot depicting the effect size of significant PRS (**A**–**F**). Upper plots, depicting the β value and the 95% CI of discrete and continuous variables, and bottom plots, depicting the odds ratio and the 95% CI of binary variables, for each regression between PRS and pigmentary traits. X axis indicates the effect size for each pigmentary trait (Y axis). Additional information about *p*-value is shown. Associations reaching the nominal significance threshold are highlighted in red.

**Table 1 genes-14-00149-t001:** Genome-wide association study of pigmentary traits results overview. Overview of the results obtained in the study, summarizing the 13 analyzed traits, type of trait (constitutive, facultative, or built), type of variable (continuous, discrete, or binary), total number of SNPs identified, number of independent SNPs, number of lead SNPs, number of risk loci, and number of structural variants.

Trait	Type	SNPs (n)	Independent SNPs (n)	Lead SNPs (n)	Genomic Risk Loci (n)	SVs (n)	λ
Red hair	Binary	181	54	28	23	4	1.0946
Blond hair	Binary	209	17	5	3	0	1.0096
Brown hair	Binary	143	22	8	4	0	1.0171
Eye color	Discrete	829	97	28	6	2	1.0101
Skin color	Discrete	478	66	20	7	2	1.0180
Freckles	Discrete	30	7	5	4	0	0.9996
Ease sunburn	Discrete	170	28	11	6	0	1.0061
Ease suntan	Discrete	190	25	9	4	2	1.0138
Sunscreen in exterior	Discrete	0	0	0	0	0	1.0167
Sunscreen when suntan	Discrete	0	0	0	0	0	1.0007
Weekly hours spent in the exterior	Continuous	0	0	0	0	0	1.0061
Phototype score	Continuous	768	111	34	6	4	1.0011
Fitzpatrick scale	Discrete	564	82	22	5	4	1.0198
Total		1515	257	99	37	8	

## Data Availability

All data is available at EGA (Dataset ID EGAD00010001665 and EGAD00010002153). GWAS summary statistics are available upon request to the authors (correspondence author: rdecid@igtp.cat), and in the GCAT|Genomes for life website form tool.

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
