# Peer review of "Skin Phototype and Disease: A Comprehensive Genetic Approach to Pigmentary Traits Pleiotropy Using PRS in the GCAT Cohort"

_genes, 2023, doi:10.3390/genes14010149_

Round 1

Reviewer 1 Report

Farré et al. conduct a GWAS on 13 pigmentary-related diseases based on  a South European cohort of ~20k individuals (the GCAT cohort), and identify 37 risk loci, 3 of which are reported to be novel, previously unreported risk-loci.

In my opinion, this is a well-powered analysis which contributes relevant genetic information on a set of phenotypic traits related to pigmentation, in a population group not sufficiently studied; i.e. Southern Europe. The manuscript is well organized, the statistical techniques are correctly applied and the inferences proposed correspond to the results obtained. I am inclined to recommend to accept the manuscript for publication, but there are a series of points I would like the authors to deal with beforehand.

1.-In the introduction, the use of "constitutive" and "facultative" does not seem too correct to me, as skin color for instance has always been described as being constitutive (i.e. color scored at unexposed regions) or facultative (color as as a result of sun exposure). A similar situation applies for hair color and even eye color. I would suggest a different terminology.

2.-To which degree do the authors consider that the sample is reflecting the general population, and how do the authors think that is affecting the estimates of heritability and PRS reported, i.e., considering that they may not be properly reflecting the general-population disease (or phenotypes) prevalence. For instance, your sample is composed of individuals aged 40-65 years, but melanoma is in general diagnosed first at 65 (on average)

3.- Although the authors include age and sex as covariates, wouldn´t it be more interesting to  analyze genetic association independently for males and females?

4.- In section 2.5, a total of 16,967,118 SNVs are reported, but in section 2.3, this figure seesm to tbe 10,195,351 SNVs (I assume that in both cases this figures are after imputatiion). Which one is  the correct one?

5.- In section 2.5, could the authors describe the full PLINK commands used? Why p2=1?

6.- In different places, it is not the first 10 (or any other number) "PCA", rather it should be the first 10 (or any other number) "PCs"

7.- Table 1 (and throughout the mansucript): why the authors consider red hair, blond hair and brown hair as 3 different traits and not just one single trait, i.e. hair color (as the authors have done for eye color and skin color)?

8.- Section 3.3: the SE for h2 seems to be quite big in some traits. Could the authors discuss this? Also, red-hair h2 is 1.916, is that correct? Shouldn´t heritability range between 0 and 1? (also applies to other regression values below)

9.- Inset text for figures 5 and 6 is not readable (too small).

10.- In the Discussion section: I do not really think that RP11-1084J3.4 is a novel pigmentation-associated gene. It rather, as the authors say, represents a naturally occurring read-through transcription between C1QTNF3 and AMACR. So the number of novel genes should be two rather than 3.

11.- Also in Discussion, there seems to be a contradiction with "our study incorporates large structural variants" and  "We have identified 15 SVs, mostly small deletions". Please explain.

12. Discussion: please defien TAGs

Author Response

Dear colleague,

We appreciate the time and effort invested to go through the manuscript so carefully and for your very appropriate comments, which helped us to improve our work. Please, find our answers to address your concerns (our replies are in green). We have revamped our original manuscript accordingly to make the draft clearer and easier to understand and follow. We believe that your suggestions and comments have improved the work considerably, and would hope that this will satisfy your inquiries.

Reviewer 1:

Farré et al. conduct a GWAS on 13 pigmentary-related diseases based on a South European cohort of ~20k individuals (the GCAT cohort), and identify 37 risk loci, 3 of which are reported to be novel, previously unreported risk-loci.

In my opinion, this is a well-powered analysis which contributes relevant genetic information on a set of phenotypic traits related to pigmentation, in a population group not sufficiently studied; i.e. Southern Europe. The manuscript is well organized, the statistical techniques are correctly applied and the inferences proposed correspond to the results obtained. I am inclined to recommend to accept the manuscript for publication, but there are a series of points I would like the authors to deal with beforehand.

1.-In the introduction, the use of "constitutive" and "facultative" does not seem too correct to me, as skin color for instance has always been described as being constitutive (i.e. color scored at unexposed regions) or facultative (color as as a result of sun exposure). A similar situation applies for hair color and even eye color. I would suggest a different terminology.

Thank you for this suggestion, we agree with the need to use a precise terminology. Therefore, we have adapted the terms used (line 39), and Table1.

2.-To which degree do the authors consider that the sample is reflecting the general population, and how do the authors think that is affecting the estimates of heritability and PRS reported, i.e., considering that they may not be properly reflecting the general-population disease (or phenotypes) prevalence. For instance, your sample is composed of individuals aged 40-65 years, but melanoma is in general diagnosed first at 65 (on average)

Thank you for pointing this out. The GCAT cohort is a population-based cohort, made for citizens from Catalonia. Complete protocol for the recruitment design and representativeness had been summarized in the paper by Obon-Santacana et al, 2018 (doi: 10.1136/bmjopen-2017-018324) (37). Yet to be published data (Blay et al, 2023, manuscript in preparation) on comparison of the cohort disease prevalence and the national public healthcare registers from the general population in Catalonia (i.e. two million registers with the same age range), indicates a bias for a healthier population and a higher socioeconomic status in the cohort. In this kind of participative study, a similar bias is always present even, as is the case of UKBB (Fry et al. 2017, doi: 10.1093/aje/kwx246), despite their original effort to made a representative cohort of the British population. However, this does not undermine the results, since the present study does not aim to translate the results to the overall population, but underscore the relationship among diseases present in a well-characterized sample, identifying complex patterns and highlighting possible common pathway in the disease incidence.

To be clearer on this point, we have added the following sentence in the main text, line 548: “However, the healthy bias of the GCAT cohort could limit the generalization to the general population”.

3.- Although the authors include age and sex as covariates, wouldn´t it be more interesting to analyze genetic association independently for males and females?

We agree that an additional stratified analysis should identify additional relationships due to the gender-specific characteristics of most of the diseases. However, given the high number of traits and diseases analyzed in this study, and the size of the genomic cohort (n=5,000), gender stratification might result in underpowered analyses. To overcome this complexity, we looked for more general patterns, common to both sexes.

4.- In section 2.5, a total of 16,967,118 SNVs are reported, but in section 2.3, this figure seem to be 10,195,351 SNVs (I assume that in both cases this figures are after imputation). Which one is the correct one?

Thank you for detecting this mistake. Both figures are after imputation, the second number, refers to the final retained variants after quality control, as indicated in section 2.3. The final core set included a total of 10,216,971 variants with MAF ≥ 0.01 and INFO score ≥ 0.7 (21,620 SVs, and 10,195,351 SNVs and small indels). All data is available at EGA (Study ID EGAS00001003018).

We have corrected the numbers in line 140.

5.- In section 2.5, could the authors describe the full PLINK commands used? Why p2=1?

We set the parameter p2 to 1 while clumping to get a detailed list of all the SNPs clumped under a lead SNP independently of its p-value. Since we agree that this might generate confusion while not being a compulsory parameter to reproduce the analysis, we have removed it from the manuscript to avoid future confusions. Additionally, we detected the p-value threshold annotated in the manuscript was not the one that was used in the analysis and fixed it to the standard genome-wide significance threshold 5e-8. 

6.- In different places, it is not the first 10 (or any other number) "PCA", rather it should be the first 10 (or any other number) "PCs"

Thank you. We have corrected it in the manuscript.

7.- Table 1 (and throughout the mansucript): why the authors consider red hair, blond hair and brown hair as 3 different traits and not just one single trait, i.e. hair color (as the authors have done for eye color and skin color)?

We agree with the reviewer that this point is not clear enough. We have included a statement in the section 2.5, text (line 139) clarifying this point.

Red hair was analyzed separately due to its differential genetic architecture, that is more similar to that of a mendelian trait, being associated with genomic variation in the MC1R and modified by few additional loci. Other hair color traits are considered as a continuum ranging from blond to black, but previous studies have unveiled some loci that appear to influence blond hair but not brown hair (Lona-Durazo et al, 2021, DOI: DOI: 10.1038/s42003-021-02764-0). According to this information, we decide to analyze the hair color separately.

8.- Section 3.3: the SE for h2 seems to be quite big in some traits. Could the authors discuss this? Also, red-hair h2 is 1.916, is that correct? Shouldn´t heritability range between 0 and 1? (also applies to other regression values below)

We agree with the reviewer's assessment, that it was not clear enough in the text.

The standard error of heritability and genetic correlation estimates are highly influences by the sample size. While heritability estimates should range from 0 to 1, in LDSC you can get negative values and values above 1. A negative estimate is not meaningful and it would hint that the true heritability is closer to 0, in our case an example of this would be Ease of suntan (h2: -0.055, se: 0.0858), while a heritability above 1 could appear due to many reasons, such as studying a highly heritable polygenic trait, or specifying a wrong number of cases in the input file. In our case, give the high standard error we observed, we believe this is due to the low number of individuals with red hair in our analysis, resulting in unreliable heritability estimates.

To further validate this, alternatively we computed the heritability of red hair using the software GCTA, which yielded a non-significant heritability estimate (h2: 1.9e-5, se: 1.74, p: 0.5). We have further clarified that the heritability estimates for red hair are unreliable due to the low number of cases in the line 265 in results, and we have added this low number of individuals with red hair in our analysis as a limitation in line 553.

9.- Inset text for figures 5 and 6 is not readable (too small).

As suggested by the reviewer, we have revised all the figures to improve its readability.

10.- In the Discussion section: I do not really think that RP11-1084J3.4 is a novel pigmentation-associated gene. It rather, as the authors say, represents a naturally occurring read-through transcription between C1QTNF3 and AMACR. So the number of novel genes should be two rather than 3.

Thank you for this observation. We have changed the former statement in the abstract and in the discussion (line 440), we have put in context the finding of this “third” gene, a read-through (RP11-1084J3.4), as a putative regulator element in a complex regulatory locus, C1QTNF3-AMACR adding a new bibliographic reference (63).

In addition, we have also actualized in the main text and supplementary material, the nomenclature for RP11-1084J3.4 and C17orf112 for the currently accepted name C1QTNF3-AMACR and LINC02876 respectively.

11.- Also in Discussion, there seems to be a contradiction with "our study incorporates large structural variants" and  "We have identified 15 SVs, mostly small deletions". Please explain.

We agree that this paragraph seems to be a little bit confusing, To avoid misleading interpretations, we have rephrased the sentence (line 466).

In brief, our analysis incorporates a large number of SVs derived from imputation using the GCAT|Panel (Valls-Margarit, et al, 2022, doi: 10.1093/nar/gkac076), and the downstream GWAS analysis identify 15 SVs associated with the phototype traits.

  1. Discussion: please defien TAGs

Thank you so much for catching this error, which we have now corrected (line 535).

We would like to thank your time reviewing this submission and we look forward to hearing from you during the reviewing process of this manuscript.

Yours sincerely,

Rafael de Cid

GCAT Chief Scientist

Genomes for Life

Germans Trias i Pujol Research Institute (IGTP)
Carretera de Can Ruti. Camí de les Escoles s/n
08916 Barcelona - Spain
Tel. (+34) 93 033 05 42

Fax: (+34) 93 465 1472

E-mail: rdecid@igtp.cat

www.genomesforlife.com

Reviewer 2 Report

The manuscript titled “Skin Phototype and Disease. A Comprehensive Genetic Approach to Pigmentary Traits Pleiotropy using PRS in the GCAT Cohort” reported the relationship between the wide range of human pigmentation traits and human genetics. The date is valuable which will be public interested.  Although most of the results looks interesting and reasonable, I have some concerns need to be addressed.

Major:

1.     The authors didn’t mention the control of confounding in the association test, there should be the population stratification given skin color, hair color and eye color difference. The relatedness of the samples was also unknown.  

2.     The authors contradict themselves in the methods of genome wide association test. In the manuscript, they used the PLINK to run the association test, however, in the supplementary, they mentioned the generalized linear model and mixed model.  As mentioned in 1, given the population stratification, the results would have some inflation if running by PLINK (simple linear regression or logistic regression).     From the QQ plot, it seems at least the “red hair” showed large inflation, other traits also showed some extent of inflation.  Note, the case for the “red hair” is few, case:control is imbalanced, logistic regression is known to be inflated.  The running by SAIGE or fastGWA-GLMM is suggested to correct the inflation issue.  For continues traits, the running of BOLT-LMM or fastGWA is suggested.

3.     The association test for eye color and skin color looks strange, as the code for the color was unknown, also the values assigned to the color was also ambiguous, one can’t say red color had a larger value than black color or vice versa.  The reason for the choice of case, control in the hair color comparison was also unknown.

4.     When calculating the heritability of binary trait , the authors can’t assume the sample prevalence is equal to population prevalence, if no prevalence data provided.

5.     After re-running of the GWAS, a further revision in the downstream analysis is needed, as this may change the results and conclusion (some still holds I believe).  I pick up one example, the heritability estimation from LDSC, the normal range for h2 is 0 ~ 1; normal range for rg is -1 to 1. Out of range value usually means the upstream association test is not performed properly (e.g. red hair, h2 = 1.96…).

Minor:

1.     GWAS and PRS was first time shown in the abstract, hence difficult for readers to know what the meaning is. A full name in the first place would be helpful for readers.

2.     The method to obtain the PGS shall be detailed.  The authors were using PLINK, hence the method shall be the C+PT, some details, e.g., the P-value cutoff, clumping parameters shall be noted in the text.  

3.     A share of the full summary statistics is suggested in GWAS studies instead of the significant locus only.

4.     The authors mentioned 16 fine-mapped genes in abstract, however, the results are not clear about this part. How is the fine-mapping are performed and what the results.

Author Response

Dear colleague,

We appreciate the time and effort invested to go through the manuscript so carefully and for your very appropriate comments, which helped us to improve our work. Please, find our answers to address your concerns (our replies are in green). We have revamped our original manuscript accordingly to make the draft clearer and easier to understand and follow. We believe that your suggestions and comments have improved the work considerably, and would hope that this will satisfy your inquiries.

Reviewer 2.

Comments and Suggestions for Authors

The manuscript titled “Skin Phototype and Disease. A Comprehensive Genetic Approach to Pigmentary Traits Pleiotropy using PRS in the GCAT Cohort” reported the relationship between the wide range of human pigmentation traits and human genetics. The date is valuable which will be public interested.  Although most of the results looks interesting and reasonable, I have some concerns need to be addressed.

Major:

1.-The authors didn’t mention the control of confounding in the association test, there should be the population stratification given skin color, hair color and eye color difference. The relatedness of the samples was also unknown.  

We appreciate this comment and we want to clarify this point to the reviewer, making clearer to the reader the quality and of the genetic dataset. 

In this study we restricted the analysis to the unrelated participants with Caucasian ancestry born in Spain (n=4,988), to mitigate bias in the genomic analysis (Galvan-Femenía et al 2018 for detailed quality control used). In addition, PCs were included in the association analysis to control for this.

In brief, population stratification and relatedness analysis were performed using agnostic methods using genomic information (Galvan-Femenía el al 2018, Galvan-Femenía et al 2020, Valls-Margarit et al 2022) identifying hidden structure relationships (i.e. population structure and relatedness). All the information required about the population and description of the GCAT sample is reported in Obón-Santacana et al, 2018 and Galván-Femenía et al 2018, and the references 37 and 39 included in the main text. 

The description of quality checks performed for population stratification is already described in the main text, line 116, and the control for confounding stratification is explained in line 143.

To be clearer on the point of cryptic relatedness we have rephrased and included this in line 116.

2.-The authors contradict themselves in the methods of genome wide association test. In the manuscript, they used the PLINK to run the association test, however, in the supplementary, they mentioned the generalized linear model and mixed model.  As mentioned in 1, given the population stratification, the results would have some inflation if running by PLINK (simple linear regression or logistic regression).     From the QQ plot, it seems at least the “red hair” showed large inflation, other traits also showed some extent of inflation.  Note, the case for the “red hair” is few, case:control is imbalanced, logistic regression is known to be inflated.  The running by SAIGE or fastGWA-GLMM is suggested to correct the inflation issue.  For continues traits, the running of BOLT-LMM or fastGWA is suggested.

Thank you for this suggestion, we agree with the reviewer that hidden stratification is a common problem in GWAS studies. However, in this case of our study (see response 1), it is clear that these concerns have been considered. We agree that this point is not clear enough so we add additional information about the genomic inflation of association test.

In our analysis is a common practice to mitigate the hidden stratification problem, defining a homogenous cohort. Previous analysis (Galván-Femenía, et al 2018) defined a sample of 4,988 unrelated participants with Iberian ethnicity, determined by self-described ethnicity, and supervised ancestry inference by Principal Components Analysis (PCA) that avoids this generalized problem. In this way, we have controlled this bias and the derived inflation for the association test by using a QCed genomic cohort. This, results in a homogenous Caucasian-ancestry-born-in-Spain cohort of unrelated cases (Galvan-Femenía et al 2018, Galvan-Femenía et al 2020, Valls-Margarit et al 2022). 

Regarding the observed QQ plot inflation in red hair color trait (l=1.09), this is an expected result from the few numbers of red-hair individuals present in the GCAT cohort (as is expected in South-European populations), and the unbalanced case-control ratio of the GWAS analysis.  We retained it in the analysis, always having in mind this low-frequency effect, because red hair color is one of the more well-reported traits to be associated with comorbidities, and widely reported in other Northern-Europe-based studies.

To allow a better interpretation of the results, genomic inflation value (l) is now provided in Table 1, for all pigmentary traits tests. In addition, we have noted this limitation in the final section of the discussion.

Additionally, the analysis that yielded the data represented in the supplementary figure 3, mentioned by the reviewer, is the result of the association analysis between the Fitzpatrick scale and data obtained from EHR. This analysis was carried out without taking into account genetic information, since it is not available for all individuals. This was done using a generalized linear model, not a mixed model, and we modified the caption of the supplementary figure accordingly. The methodology for this analysis is mentioned in section 2.4. 

3.-The association test for eye color and skin color looks strange, as the code for the color was unknown, also the values assigned to the color was also ambiguous, one can’t say red color had a larger value than black color or vice versa.  The reason for the choice of case, control in the hair color comparison was also unknown.

We apologize that this has not been clear enough in the manuscript.

As required for the reviewer, we have made changes in the main text, to allow a clearer interpretation of the results.

Regarding code color used, in section 2.2, line 92, we have defined the code for the color and case-control assignation. Additionally, it is also defined in the supplementary material table 1 (cases and controls), and 2 (phototype assessment).

About the choice of a separate analysis for hair color. In section 2.5, line 139, we have now added a justification of the separated analysis of red hair, blonde and brown color due to different expected genetic basis, being mendelian inheritance pattern predominant in the red color hair trait (Lona-Durazo et al, 2021, DOI: DOI: 10.1038/s42003-021-02764-0).

4.-When calculating the heritability of binary trait , the authors can’t assume the sample prevalence is equal to population prevalence, if no prevalence data provided.

We agree on this point. The reason is due to the scarcity of data available in the Iberian population, and their controversial reported data on control samples, Gomez-Acebo et al 2018 (25), that move us to consider sample and population prevalence as equal for this analysis.

5.- After re-running of the GWAS, a further revision in the downstream analysis is needed, as this may change the results and conclusion (some still holds I believe).  I pick up one example, the heritability estimation from LDSC, the normal range for h2 is 0 ~ 1; normal range for rg is -1 to 1. Out of range value usually means the upstream association test is not performed properly (e.g. red hair, h2 = 1.96…).

While we appreciate the reviewer’s feedback, we have not re-run the whole analysis because we understand that initial concerns of the reviewer regarding the impact of population stratification and relatedness bias are now well addressed (responses 1 and 2). We have included in the manuscript the details of the pre-treatment of the analyzed sample and the measures used to control for confounding in the GWAS analysis.  Genomic inflation values (l) are now provided in the manuscript for each pigmentary trait GWAS, being near l=1.1 in the case of red hair color trait. This has been commented in the main text to justify the abnormal values observed in heritability observed in this trait.

Minor:

1.-GWAS and PRS was first time shown in the abstract, hence difficult for readers to know what the meaning is. A full name in the first place would be helpful for readers.

We thank the reviewer for the comment and changed the text accordingly, avoiding the use of acronyms in the abstract.

2.-The method to obtain the PGS shall be detailed.  The authors were using PLINK, hence the method shall be the C+PT, some details, e.g., the P-value cutoff, clumping parameters shall be noted in the text.  

In our analysis we did not derive any PGS, SNP weights from already computed PGS were obtained from the PGS Catalog. We described in detail this information in line 186: “We used weights for relevant diseases from the PGS Catalog [50]”. Also, PGS-ID from the Polygenic Catalog and source reference is provided in Supplementary Table 4.

3.-A share of the full summary statistics is suggested in GWAS studies instead of the significant locus only.

We agree with the reviewer’s assessment. The summary statistics of the GWAS are available upon request to the correspondent author (rdecid@igtp.cat), and in the GCAT|Genomes for life repository. The link to the online form is provided in the Data Availability Statement paragraph (line 609). We will process the requests and send a link to our repository.

3.-The authors mentioned 16 fine-mapped genes in abstract, however, the results are not clear about this part. How is the fine-mapping are performed and what the results.

Thank you for this suggestion. In our study we refer to fine-mapping as the functional annotation of the GWAS hits however, we agreed that fine-mapping term may be misleading, and we remove it accordingly, in the abstract and graphical abstract.

We would like to thank your time reviewing this submission and we look forward to hearing from you during the reviewing process of this manuscript.

Yours sincerely,

Rafael de Cid

GCAT Chief Scientist

Genomes for Life

Germans Trias i Pujol Research Institute (IGTP)
Carretera de Can Ruti. Camí de les Escoles s/n
08916 Barcelona - Spain
Tel. (+34) 93 033 05 42

Fax: (+34) 93 465 1472

E-mail: rdecid@igtp.cat

www.genomesforlife.com

Round 2

Reviewer 2 Report

The authors addressed my concern well.   One of my concerns about the population stratification, the mixed model are still needed if the authors would like to reduce the false postive rate (some traits would have reasonable results), even though using the unrelated sample only. The population factors were still there, although they were all sampled from same cohort, they were likely in different sub-population.   However, I accept the current version as the story told is also complete and reasonable, and hope the authors validate the results in their future study.